# Land tenure regimes influenced long-term restoration gains and reversals across Brazil's Atlantic forest

Rayna Benzeev [1,2] ✉, Sam Zhang[3,4,5], Pedro Ribeiro Piffer[6] & Megan Mills-Novoa[1,2]

Forest restoration is increasingly promoted to mitigate climate change, conserve biodiversity, and secure food and water sovereignty. Yet many restored forests do not persist in the long term, and the role of land tenure regimes in shaping these outcomes remains poorly understood. We examine restoration reversals (restored forests later deforested) and long-term restoration gains (restored forests that remained intact) across 1.9 million territories in Brazil's Atlantic Forest from 1985 to 2022. We compare Indigenous lands, Afro-descendant (*Quilombola*) territories, agrarian-reform settlements, protected areas, and private properties, introducing a statistical matching technique—*agglomerative matching*—to account for systematic differences between land tenure regimes. We find that Indigenous lands and agrarian-reform settlements exhibit significantly more long-term restoration gains than private properties. Concurrently, and by a smaller margin and on a smaller land area, Indigenous lands and agrarian-reform settlements exhibit higher reversals. These results highlight the relatively low restoration longevity of private properties and emphasize the importance of socio-political conditions in enabling long-term restoration gains.

Across the globe, over 70 countries have committed to restoring an aggregate of 350 million hectares of forest by 2030. The UN General Assembly has declared the current decade (2021–2030) the Decade on Ecosystem Restoration, indicating a dedication to restore the planet[1]. Enacting these pledges is expected to produce positive impacts by drawing down carbon to slow climate change, protecting endangered species and biodiversity, alleviating poverty through job creation, maintaining healthy watersheds, and securing food sovereignty for local people[2–6]. However, many restoration planners and investors have not assessed the extent to which restoration has persisted in order to successfully achieve these proclaimed positive impacts. When evaluating forest restoration (hereafter restoration)—defined as forest gain, including any change from productive land uses to natural forest cover[7]—one critical distinction is the difference between long-term restoration gains—restored forests that remained intact in the long term (defined here as restoration that persisted for a minimum of 10 years and until the end of the measured time period (2022)), and restoration reversals—restored forests later deforested (defined here as restoration that persisted between 3 and 10 years and was deforested before the end of the measured time period—see "Methods"). Restoration reversals have been shown to be ten times more prevalent than long-term restoration gains[8], restoration has often accounted for only a small carbon sink[9], restoration has not often contributed to large net forest gains[10,11], and many restoration projects (e.g., planned initiatives to intentionally plant and restore forests) have not achieved large increases in forest cover[12]. As such, there is a strong need for

[1]Department of Environmental Sciences, Policy, and Management, University of California, Berkeley, CA, USA. [2]Energy and Resources Group, University of California, Berkeley, CA, USA. [3]Santa Fe Institute, Santa Fe, NM, USA. [4]Department of Mathematics & Statistics, University of Vermont, Burlington, VT, USA. [5]Vermont Complex Systems Institute, University of Vermont, Burlington, VT, USA. [6]Department of Ecology, Evolution and Environmental Biology, Columbia University, New York, NY, USA. ✉e-mail: rayna.benzeev@gmail.com

evidence on the conditions that enable restoration gains in the long term.

Many restoration projects have failed to achieve their stated objectives[12–14], and many of the proximate causes of failures of restoration projects have been social and political[15–17]. Some of the most important socio-political enabling and/or constraining factors have been land tenure, livelihoods, governance arrangements, and monitoring and enforcement[18,19]. However, a lack of consistent data at scale has made it difficult to analyze how social and political factors influence restoration outcomes. One socio-political variable that intersects with several of the social enablers/constraints of restoration is land tenure regime–defined as "the combination of tenure-related governance factors that exist over a given parcel of land and are stable over a certain period of time"[20(p. 1)]. These governance factors determine (a) land tenure–the formal bundle of powers granted to a territory (e.g., access, use rights, management, and exclusion)[21,22], and (b) tenure security (e.g., the content of rights, their duration, and their robustness)[23,24]. The differences in the powers granted through formal/informal land tenure and tenure security/insecurity result in inherent differences in land tenure regimes–such as private properties, communal territories, protected areas, and public lands–which consequently influence environmental outcomes[20,25–28].

An important remaining question is whether land tenure regimes have influenced restoration. Although land tenure regime has been an important determinant of deforestation[20], restoration occurs through its own set of unique socio-environmental and biophysical processes. One difference in restoration compared to deforestation is that much restoration has occurred through natural regeneration[29–32]–a process where degraded and abandoned agricultural land regenerates naturally into forest either intentionally or unintentionally[33]. Although many active restoration projects exist where people focus on physical tree planting, natural regeneration has been more common across many geographies, particularly in locations where a large percentage of forests remain standing[29–31,34]. Importantly, both active restoration and natural regeneration have been considered forest restoration (according to our definition), since both have resulted in substantial forest gain[7,35–40]. However, natural regeneration has resulted in high rates of restoration reversals, partially because unintentionally restored land has been later desired as productive land[8,10,11,36,37]. Land tenure regimes may determine where people have enabled forest regrowth, prevented the deforestation of regrown land, and/or abandoned land in the long term. However, we know of no study that has examined how land tenure regimes influence restoration outcomes, and certainly not for specific types of restoration outcomes (reversals and long-term gains).

As a country that contains a third of the world's tropical forests and rich socio-cultural diversity, Brazil is a key context to evaluate the role of land tenure regimes for restoration reversals and long-term gains. Within Brazil, the Atlantic Forest biome is a top priority ecosystem for restoration[41]. While the Brazilian Amazon is the primary focus for initiatives to reduce deforestation given that 80% of its original forest cover is still remaining[42], the Atlantic Forest is a focus for restoration since only 28% of original forest cover remains[43]. Large-scale analyses of restoration in the Atlantic Forest have used measures of forest gain as a determinant of restored ecosystem services and biodiversity recovery[31,35]. Research investigating restoration outcomes in the Atlantic Forest has found a higher probability of long-term restoration gains in areas with steeper slopes, higher GDP, higher agricultural yields, and that have proximity to rivers, existing forests, and contain mechanized agriculture[37]. Although restoration has occurred more frequently within agricultural landscapes, it has also been more likely to be reversed in these landscapes[37]. Gaining an understanding of where restoration reversals and long-term gains have occurred is incomplete without understanding the socio-political

context of the land tenure regime in which these reversals and long-term gains have taken place.

Brazil's socio-political landscape includes a mix of communal and private land tenure regimes, such as private properties, Indigenous lands, agrarian reform settlements (*assentamentos*), Afro-descendant (*Quilombola*) territories, protected areas, and public lands (Table 1). Although many land tenure regimes fall somewhere along a communal-private spectrum of rights arrangements and territorial zoning characteristics, we consider the land tenure regimes described in Table 1 as relatively communal, with the exception of private properties (and certain protected areas that prohibit human settlement–see Table 1). Each land tenure regime encompasses a range of possibilities for formal or informal land tenure (as recognized by the State), tenure security, and individual or collective use rights (Table 1). Formalization varies greatly by land tenure regime in Brazil; some territories are formalized from the start, while others require an arduous multi-step process to gain formality[44,45]. Although secure tenure is more difficult to measure compared to formalization–given that formalized recognition of rights does not guarantee protection of rights or access to rights–tenure has often been considered secure when a "legal protection against forced evictions" is guaranteed over time[24,46].

The largest transition in land tenure regimes that occurred alongside colonization (by various waves of settlers over the past several hundred years) was privatization[20]. Although this privatization has largely stabilized, lands with weak tenure security from any land tenure regime are still under threat of being claimed by squatters, who are sometimes able to obtain titles for private ownership[47]. Understanding the environmental implications of broader land tenure transitions from communal territories to private properties requires an examination of how land tenure regimes shape restoration. Determining how restoration outcomes compare between land tenure regimes is a critical first step in understanding this broader relationship. In addition, most programs and policies on restoration in Brazil influence private properties, and most planned restoration initiatives in Brazil have taken place in private properties. For example, Brazil's Forest Code mandates that all private properties in the Atlantic Forest maintain 20% of their property forested and for noncompliant landowners to conduct restoration. Other programs, such as the Atlantic Forest law, increased monitoring and enforcement, the Atlantic Forest Restoration Pact, voluntary carbon offset markets, and other initiatives aimed at achieving Brazil's Nationally Determined Contribution to mitigate climate change have also principally targeted restoration of private properties. Although private properties have the largest restoration debt due to their substantial contribution to deforestation, and some programs should not be applied to other land tenure regimes due to their social consequences[14,48,49], only focusing restoration initiatives in private properties prevents restoration financing from reaching these other land tenure regimes. No study has compared how restoration in private properties compares to differential restoration outcomes in other land tenure regimes.

We developed a technique within the field of statistical matching– agglomerative matching–to analyze how land tenure regimes influenced restoration reversals and long-term gains in the Brazilian Atlantic Forest from 1985 to 2022. This technique iteratively combines multiple private properties into groups of private agglomerates to achieve covariate balance when matching treatment and control territories (see "Methods" and SI for full details). We matched territories with similar distributions of covariates and then conducted linear modeling to isolate the persistent structural differences across land tenure regimes on restoration, measuring restoration that occurred through both active restoration and natural regeneration. Our approach overcomes past challenges for land tenure regimes that have systematically different spatial and biophysical characteristics, enabling us to avoid dropping any data and retain the full sample of all

**Table 1 | Territorial characteristics and governance arrangements of land tenure regimes: private properties, Indigenous lands, assentamentos, Quilombola territories, and protected areas in the Atlantic Forest**

| Land tenure regime | Definition | Range of formal to informal tenure rights | Range of tenure security to insecurity | Range of communal to private ownership | Amount of total land in the dataset (ha) | Number of lands in the dataset |
|---|---|---|---|---|---|---|
| Private properties | Private lands registered in Brazil's Rural Environmental Registry (Portuguese acronym: CAR) of property boundaries | Formal (in this study, although informal private properties exist outside of this study) | Secure tenure may be obtained with title. However, tenure may be impermanent with transfer of titles | Private | 67,820,759 | 1,884,269 |
| Indigenous lands | Original rights to lands traditionally occupied by Indigenous Peoples, which are mandated to be demarcated, protected, and respected according to the Brazilian Constitution (Article 231) | Informal until demarcated, and then formalized. Currently, 67% of Indigenous lands in Brazil have formal tenure[100] | Four-stage process to obtain permanent and secure demarcation, but insecure until demarcated | Communal title designated by law | 656,842 | 143 |
| Agrarian-reform settlements (assentamentos) | Settlements designated by the National Institute of Colonization and Agrarian Reform (INCRA) to provide agricultural land and communal infrastructure to formerly landless farmers | Formal | Secure. Non-negotiable sale of title for a period of 10 years (Article 189) | Communal title (but can be parceled by individual ownership) | 1,428,905 | 1408 |
| Quilombola territories | Territories of ethnic-racial groups identified based on self-definition, with distinct territorial relations, and Black ancestry related to resistance to historical oppression (Decree 4887/03)[101], with collective ownership recognized in the Brazilian Constitution (Article 68) | Informal until regularized, and then formalized. Currently, only approximately 3% of Quilombola territories in Brazil have formal land tenure[83] | Six-stage process to obtain permanent, secure regularization, but insecure until regularized | Communal title designated by law | 154,575 | 100 |
| Protected areas | Includes areas of (a) strict protection—with a primary objective of biodiversity conservation that prohibits human settlement and extraction, and (b) sustainable use—which permits the sustainable use of natural resources and allows human settlement | Formal | Secure unless downgrading, downsizing or degazettement is implemented by law | (a) Strict protection: controlled by environmental agencies, (b) Sustainable use: collective management | 2,871,611 | 697 |

Public lands are also a land tenure regime in Brazil, but were not included since they do not appear in the analysis (see "Results", "Methods", and SI). See SI to visualize land areas and numbers of lands (Fig. S4).

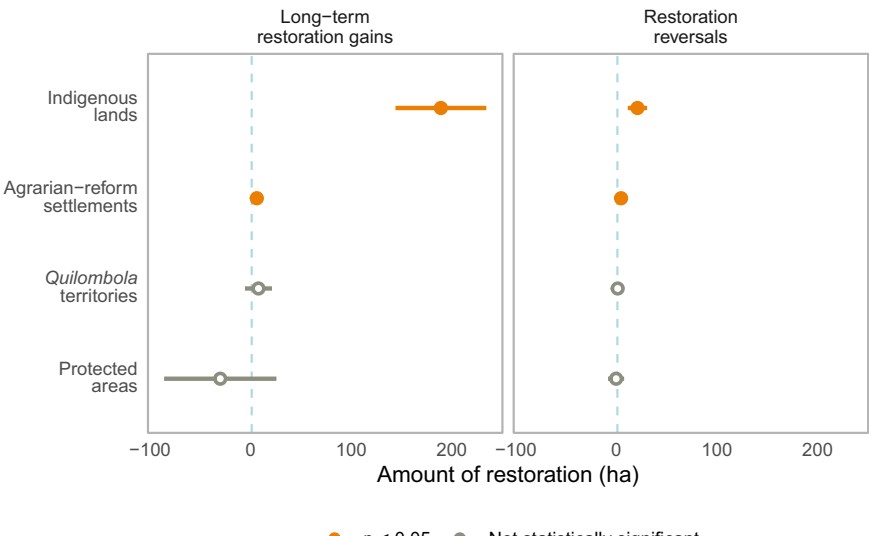

**Fig. 1 | Estimates of regression models after matching for restoration reversals and long-term restoration gains.** All models were matched with private properties. Error bars indicate one standard error of the mean. Orange bars indicate statistical significance and gray bars indicate a lack of statistical significance at

$p < 0.05$. Effect sizes were fit based on treatment condition and cannot be compared between models. Sample sizes of each condition are available in Table 1, and exact estimates and $p$-values are presented in Table S1.

Indigenous lands, agrarian-reform settlements, *Quilombola* territories, and protected areas. We integrated a comprehensive dataset of 1.9 million territories in the Atlantic Forest[50] with a dataset of pre-processed land use change data[51] to compare the effects of restoration reversals and long-term gains in Indigenous lands, agrarian-reform settlements, *Quilombola* territoriess, and protected areas to effects in private properties. We defined restoration reversals as restoration that persisted between 3 and 10 years but that was deforested before the end of the time period (2022), and long-term restoration gains as restoration that persisted for a minimum of 10 years and until the end of the time period (2022) (see "Methods"). By characterizing how restoration reversals and long-term gains varied between land tenure regimes, our study contributes to understanding how transitions in land tenure regimes, such as through privatization, land grabs, demarcation, territorial defense, and occupation of ancestral lands, could influence the future of restored forests.

## Results and discussion

### Differences between land tenure regimes

We found that Indigenous lands and agrarian-reform settlements had significantly more long-term restoration gains than private properties, while protected areas and *Quilombola* territories had no significant differences. Specifically, Indigenous lands had an average of 189 ha more long-term restoration gains than private properties, and agrarian-reform settlements had an average of 6.49 ha more long-term restoration gains than private properties (Fig. 1 and Table S1). In relation to restoration reversals, we found that Indigenous lands had an average of 21 ha more restoration reversals than private properties, and agrarian-reform settlements had an average of 4.47 ha more restoration reversals than private properties (Fig. 1 and Table S1). *Quilombola* territories and protected areas did not have significant differences in restoration reversals compared to private properties. See SI to visualize the variability of model covariates (Fig. S1).

### Model robustness

We ran each model 500 times to compare standardized mean differences (SMD) and prognostic scores across iterations (Table S3). The models reported in Table 1 represent the model from the 500 model

iterations that had the minimal distance between covariates—i.e., the most similar covariate distributions between treatment and control groups—after agglomerative matching. SMD values remained within the ±0.1 range for all treatment conditions, with the exception of private properties, which had SMD values > 0.1, meaning that private properties were too dissimilar to private properties to find sufficient matches, and thus were only included and discussed in the SI (Table S3). We additionally confirmed that each best-fitting model had similar effect sizes to the other 500 runs of agglomerative matching (see "Methods" and Table S3).

### Descriptive statistics of long-term restoration gains and reversals

Biome-level areas of long-term restoration gains exceeded restoration reversals across all land tenure regimes in the Atlantic Forest in our dataset (2,253,850 ha of long-term restoration gains and 459,699 ha of restoration reversals from 1985 to 2022). The total percentage of land area for each land tenure regime that became a long-term restoration gain during the study period was 5.43% for Indigenous lands, 4.74% for *Quilombola* territories, 4.19% for agrarian-reform settlements, 3.07% for private properties, and 2.37% for protected areas. The total percentage of land area for each land tenure regime that became a restoration reversal was 1.13% for agrarian-reform settlements, 0.91% for *Quilombola* territories, 0.64% for private properties, 0.56% for Indigenous lands, and 0.21% for protected areas.

### Discussion of key results

Our analysis demonstrates that distributions of long-term restoration gains and reversals vary across land tenure regimes and socio-environmental landscapes. Our primary finding is that in the Atlantic Forest—the biome that has been the epicenter of restoration efforts in Brazil—Indigenous lands had exceptionally high rates of long-term restoration gains (per unit area) compared to private properties, and agrarian-reform settlements also had significant rates of long-term restoration gains. While previous studies have shown that long-term restoration in the Atlantic Forest tended to occur in areas with specific biophysical conditions (e.g., steep slopes and proximity to rivers and forests)[37], our study demonstrates that when controlling for biophysical

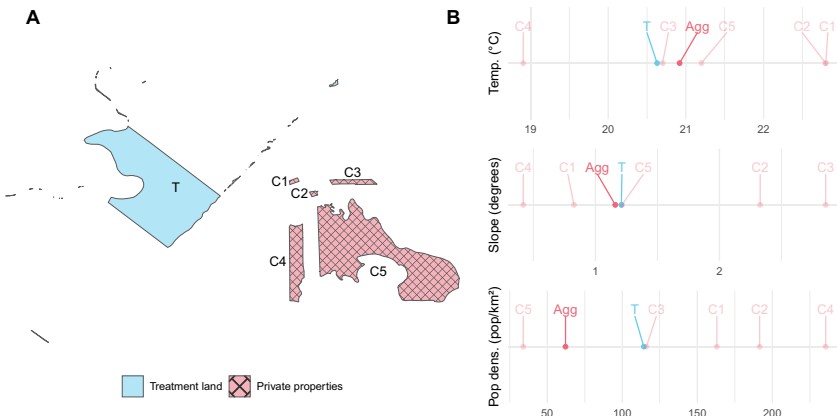

**Fig. 2 | Agglomerative matching approach. A** An example treatment land (T) matched with five control private properties (C1, C2, C3, C4, and C5) according to the distributions of ten covariates (territory size, percentage of forest cover in 1985, temperature, precipitation, slope, elevation, distance to roads, distance to rivers, distance to nearest city, and population density) (Table S2). **B** Example covariate distributions of the treatment land and control private properties for three sample covariates: temperature, slope, and population density. "Agg" represents the area-weighted average of covariate values of private properties following agglomerative matching. Control private properties were selected using the agglomerative matching algorithm (see "Methods").

conditions, long-term restoration has changed alongside socio-political context, as measured through differences in land tenure regime.

By highlighting that long-term restoration has been influenced by socio-political factors, our study demonstrates that people have enabled restoration to occur. In contrast to the passive land abandonment that has commonly enabled natural regeneration in private properties[52–54], in Indigenous lands and agrarian-reform settlements it is likely that communities have permitted trees to persist over time, either intentionally or unintentionally. Given that most restoration that has occurred in the Atlantic Forest has been natural regeneration[31], our study highlights a need to broaden the concept of restoration and the types of initiatives that are considered to be restoration to also include the interests of those communal territories that have enabled the highest rates of long-term restoration (i.e., Indigenous lands and agrarian-reform settlements).

Our results additionally indicate that Indigenous lands had significant rates of restoration reversals (although effect sizes were nine times larger for long-term restoration gains than for reversals, and the total area of long-term restoration gains was around ten times more than reversals—see SI). Agrarian-reform settlements also had high rates of restoration reversals with more similar effect sizes between reversals and long-term gains. These results correspond to the idea that restoration reversals may be inherent to socio-environmental systems, and need not detract from longer-term restoration[55].

## Underlying mechanisms of long-term restoration gains and reversals

There are six primary explanations that we believe contribute to differences in long-term restoration gains and reversals by land tenure regime, and that we hypothesize may contribute to long-term restoration gains and reversals in tropical forest restoration elsewhere. First, governance arrangements likely determine how newly restored forests are managed and continue to be managed between land tenure regimes[25,49]. For example, agrarian-reform settlements tend to have strong cooperative collective dynamics[56] that facilitate a sense of belonging[57] and participatory democracy[58], which in turn may assist with common pool resource management[59]. Second, different land tenure regimes have different possibilities for territorial zoning, influencing how land is parceled and how much land is available for restoration within a given territory. For example, Indigenous lands, agrarian-reform settlements, and *Quilombola* territories often have designated forest reserves where forests are segmented into larger

fragments, thus leaving more room where forests can regenerate compared to smaller or medium-sized private properties that may contain smaller regrowing patches of forest. Third, farming and agroecological land use practices in Indigenous lands and agrarian-reform settlements may be more extensive than private properties, or may more commonly involve leaving large plots of land to fallow and regrow to later to be deforested again, which may explain the frequent reversals[60]. Fourth, Indigenous forest cosmologies are often centered around care, reciprocity, and relational connection to forest species, which may enable restored forests in Indigenous lands to persist for longer periods of time[25,61]. Fifth, land abandonment is frequent within private properties and has resulted in restoration reversals alongside transitions in land ownership and shifts in land use strategies[36,53,54]. Sixth, land use policy often influences rates of reversals. For example, the Atlantic Forest Law (2006) in the Atlantic Forest—which prohibits deforestation of forests greater than 10 years of age—has caused many landowners to reverse restoration after 8–10 years to maintain the flexibility to use their land for non-forest purposes, and to maintain land value[36,62].

## Methodological contributions

We developed a technique within the field of statistical matching—agglomerative matching—that iteratively combines multiple private properties into private agglomerates to achieve covariate balance when matching territories (Fig. 2). Indigenous lands, agrarian-reform settlements, *Quilombola* territories, and protected areas are designated as the "treatment" groups, and private properties constitute the "control". In contrast to other matching approaches that require dropping treatment data without a sufficient match, our approach does not exclude any treatment lands. Our analysis also has the strength of being able to include many different covariates while still achieving covariate balance. Standard matching approaches may have matched treatment lands only with those private properties that had similar enough covariates (for example, the largest and most forested private properties), or may not have been able to find any adequate match for treatment lands at all. This standard form of matching would have excluded many treatment lands from the analysis (see SI—where we run matching without agglomeration). In the Atlantic Forest, excluding smaller private properties (for example) would have meant excluding the majority of properties in the biome. It would have also biased our sample of private properties to only represent a very specific type of private property (such as a larger private property, for

example). Our agglomerative matching approach therefore reduces bias by enabling a more diverse set of private properties to be analyzed, in addition to improving covariate balance between treatment and control groups—a necessary condition for matching analyses.

## Contributions to forest transition theory

Our analysis revealed that long-term restoration gains and reversals can occur concurrently within the same landscape, depending on the specific matrix of land tenure arrangements. In Indigenous lands, we found extremely high effect sizes and significant amounts of long-term restoration gains, which indicates that long-term transitions to restoration have occurred. This supports the forest transition model where farmland is thought to be able to transition to long-term restoration over time[55,63–65]. In contrast, in private properties we found that long-term forest transitions have not occurred at high rates. These trends in private properties are more characteristic of the reforestation treadmill—an idea suggesting that long-term forest transitions cannot occur due to frequent reversals[11]. For example, prior studies have found that restoration reversals were 1.5–2.3 times more likely than long-term restoration gains across the tropics[11], ten times more common across Latin America and the Caribbean[8], and that long-term restoration gains only accounted for a small carbon sink in the Brazilian Amazon[9]. In the Atlantic Forest, one third of all forests that regenerated were reversed after an average of only 8 years[37]. However, although private properties had frequent reversals, Indigenous lands and agrarian-reform settlements with similar spatial and biophysical characteristics had even higher rates of restoration reversals. The high rates of reversals across all land tenure regimes are characteristic of the "pulsed" forest transition model—which asserts that restoration reversals may be inherent to land use change rather than detrimental to the broader restoration trajectory[55]. Our study adds to a broader understanding of forest transition theory by showing that sociopolitical conditions can determine the extent to which forest transitions, treadmills, and/or pulses occur.

## Implications of respective land tenure regimes for restoration

By showing that Indigenous lands and agrarian-reform settlements had higher rates of long-term restoration gains than private properties with similar spatial and biophysical characteristics, our study corresponds to other studies that found privatization to lead to forest loss in Brazil[20,66–68]. Given that communal land tenure regimes have been disincentivized through historical public policies and pro-agribusiness decision-making under the rationale that they are "empty" or "unproductive"[69], our results at least partially contribute to evidence that some communal lands have been "productive" through long-term restoration gains, which adds to other forms of productivity, such as being economically productive through agroforestry[70], as well as socially productive by fulfilling the "social function" of property according to the Brazilian Constitution[71]. Although privatization has been the dominant trend in the Atlantic Forest in the past, efforts can still be made to slow and reverse land acquisition for private use, as well as to strengthen efforts for formalized and secure communal land tenure.

Our results support a growing body of evidence that forest outcomes in Indigenous lands have consistently outperformed other land tenure regimes. Indigenous lands have been shown to reduce deforestation compared to other land use types across the tropics[72–76], as well as to increase restoration in the Brazilian Amazon[25]. Land tenure in Indigenous lands has also improved forest cover in the Atlantic Forest[44]. Our study advances past research by showing that long-term restoration was more prevalent in Indigenous lands compared to private properties (per unit area), with a higher effect size than all other land tenure regimes. In addition, Indigenous lands had higher rates of restoration reversals than private properties, which indicates that production occurred simultaneously alongside long-term restoration.

In combination, these results suggest that avoiding land tenure transitions from Indigenous lands to other land tenure regimes could improve the efficiency of long-term restoration outcomes.

We found that both long-term restoration gains and reversals were significantly higher in agrarian-reform settlements than in private properties. Agrarian-reform settlements have been shown to contribute considerably to deforestation rates[77–79], including contributing to as much as one third of deforestation in the Brazilian Amazon[80], although deforestation rates in agrarian-reform settlements have reduced over time[81,82]. Agrarian-reform settlements additionally contribute to reducing inequities in land distribution; there are approximately five million landless families directly or indirectly demanding land across Brazil who would benefit from the establishment of agrarian-reform settlements[69]. Our results demonstrate that agrarian-reform settlements may have been able to compensate for some deforestation through rates of long-term restoration gains that were higher than those in private properties. However, agrarian-reform settlements also had higher rates of restoration reversals than private properties. Given that agrarian-reform settlements are established for the purpose of smallholder agriculture, our results may have occurred due to the high prevalence of agroforestry in agrarian-reform settlements. Agroforestry may have allowed for higher seed dispersal rates as well as mixed shade canopy to facilitate more natural regeneration—both long-term (restoration gains) and temporary (restoration reversals). Although reversals have occurred at high rates in agrarian-reform settlements, long-term gains occurred at even higher rates, demonstrating that agrarian-reform settlements contributed considerably to long-term forest restoration.

We contribute to a knowledge gap on the relationship between land tenure and forest outcomes among *Quilombolas*. Past studies have found that *Quilombola* territories had less deforestation than other land tenure regimes in Brazil[20], as well as increased restoration in the Brazilian Amazon when analyzed in combination with Indigenous lands[25]. In contrast, there has been no evidence that land tenure has influenced deforestation or restoration rates in tenured compared to non-tenured *Quilombola* territories[83]. Our study contributes to the sparse literature on the relationship between *Quilombola* territories and environmental outcomes by showing no significant difference in restoration outcomes compared to private properties. However, there is very little data available on *Quilombola* territories, and our small sample of 100 *Quilombola* territories (the highest number available from the Imaflora dataset in the Atlantic Forest with data for all our covariates) may not have been representative of the other 5972 *Quilombola* territories across Brazil[84]. If the Brazilian government were to register the thousands of *Quilombola* territories that have not been able to start the process of formalizing land claims, then there would more data available on *Quilombola* territories, it would be possible to better understand land use trends in *Quilombola* territories, and there would be a clearer path forward for *Quilombola* territories to obtain land rights[25,83].

Although protected areas have generally had deforestation rates similar to Indigenous lands[72,75,85,86], our study demonstrates that protected areas did not have significantly different amounts of long-term restoration gains and reversals compared to private properties. These results are consistent with another study, which found that protected areas in the Brazilian Amazon reduced deforestation but did not contribute greatly to restoration, particularly in comparison to Indigenous lands and *Quilombola* territories[25]. One explanation for this trend is that protected areas were already highly forested and may not have had much additional space for restoration to occur (Fig. S1). As such, they likely had similar amounts of restoration to those private properties that were also highly forested, highly remote, and with low population densities. Although protected areas may have made strong contributions to long-term restoration in the more distant past (before 1985), our results indicate that they have not contributed any more

than similar private properties to long-term restoration following 1985. While protected areas have been critical for conservation[74,87,88], their restoration contribution has been less substantial in the Atlantic Forest (per unit area).

### Broader applications for restoration decision-making and practice

Our study provides a rationale for restoration decision-makers and practitioners to reconsider how restoration is conducted and how it could become more equitable (i.e., by reducing disparate restoration impacts and increasing restoration benefit sharing). We highlight that some of the communities that have contributed most significantly to long-term restoration gains are communities who have in many cases been marginalized and excluded from formal, funded restoration projects, and who have received little (if any) funding for restoration[89]. Therefore, there is a need to more strongly recognize those communities that have been making contributions to long-term restoration for decades, as well as to support those communities that may wish to conduct more active forms of assisted natural regeneration and have not been given this opportunity in the past. Although Indigenous lands and agrarian-reform settlements cover a much smaller land area compared to private properties (Table 1), their relative contribution to long-term restoration gains has added up (there were 95,568 ha of long-term restoration gains in the Indigenous lands and agrarian-reform settlements included in our analysis from 1985 to 2022). As such, we highlight a double exclusion that many communal territories face; not only are communities with informal and/or insecure land rights being excluded from access to demarcated and formalized lands, but these communities are also being excluded from participating in certain forms of restoration. Indeed, in the case of Indigenous lands, our results demonstrate that both formal and informal lands have contributed to long-term restoration gains. However, in practice many formal restoration projects–both those that are negotiated by communities themselves as well as those that involve external partnerships and funding—require formal tenure rights to gain funding and support. As a result, Indigenous peoples without formal tenure are being excluded from the necessary conditions to determine how much and what kind of restoration they would like to conduct on their own terms. It is therefore important to (a) protect existing communal land tenure regimes from threats of land grabbing and privatization, (b) more explicitly promote conditions for natural regeneration (rather than a sole focus on active planting) in communal land tenure regimes, and (c) reconsider strengthened land and self-determination rights for communal territories as a distinct restoration strategy. Taking these considerations into account is particularly relevant for existing restoration projects that take place on lands that formerly belonged to Indigenous peoples and local communities. Although initiatives in private properties will be essential to achieving Brazil's international restoration commitments and to compensate for high rates of past deforestation, restoration that has occurred and persisted on other land tenure regimes have also contributed towards achieving these commitments.

### Limitations

There are two primary limitations of our analysis. First, there are relatively few Indigenous lands and *Quilombola* territories compared to other land tenure regimes in the Atlantic Forest ($n = 143$ Indigenous lands and $n = 100$ *Quilombola* territories), partially due to missing data (see explanation above), providing us with a smaller sample size and subsequently less robust estimates of restoration trends compared to other land tenure regimes. Second, our agglomerative matching approach does not account for potential boundary effects of private agglomerates. The larger area of borders between lands in private agglomerates may affect other factors, such as seed dispersal across borders, which could either inhibit or facilitate natural regeneration

(see SI for full details). However, we believe that this boundary effect assumption is preferable to dropping large quantities of data in cases when treatment lands have no sufficient match.

### Conclusions and directions for future research

Forest restoration is fundamentally linked to issues of land, tenure, security, governance, formalization, and rights. When restoration practitioners, decision-makers, and researchers do not recognize the interconnection of restoration with differences in land tenure regimes as part of restoration theory, decision-making, and practice, restoration may have less of a chance of persisting in the long term. Brazil is a critical context to consider socio-political restoration reversals and long-term gains, and these insights may also apply to global conversations on the longevity of restoration worldwide, including how to inform broader restoration perspectives, investigations, and action. We found that Indigenous lands and agrarian-reform settlements had significantly more long-term restoration gains than private properties, where Indigenous lands had an exceptionally large magnitude of long-term restoration gains in comparison to private properties. Indigenous lands and agrarian-reform settlements also had significantly more restoration reversals compared to private properties, although by a smaller margin for each. More research is needed to understand the mechanisms and nuances underlying the governance arrangements and other important drivers that determine the differences in restoration outcomes between land tenure regimes. In addition, more research on the impacts of large-scale land grabbing, transitions from private properties to other land tenure regimes, and ways that land claims could be strengthened when tenure formalization is not politically feasible could contribute to research in the Brazil context. With emergent land- and justice-oriented approaches to restoration becoming more prevalent, there are great opportunities for restoration initiatives to recognize and support the socio-political conditions that enable and contribute to long-term restoration gains.

## Methods
### Summary of matching approach

Our agglomerative matching algorithm iteratively sorts through private properties (see algorithm below) to select a combination of private properties with covariates similar to each Indigenous land, agrarian-reform settlement, *Quilombola* territory (QT), and protected area. Treatment lands were Indigenous lands, agrarian-reform settlements, *Quilombola* territories, or protected areas, and control lands were private properties. After agglomeration, we conducted optimal full matching (OFM), where every treated territory was matched with either one private agglomerate or one private property (Fig. 2). Using the example of Fig. 2, standard matching approaches may have matched T only with C5, or may not have been able to find any adequate match for T. Our approach also improved covariate balance; the private agglomerate ("Agg") (Fig. 2) had means for covariates temperature and population density more similar to treatment land T than private property C5. For a more extensive comparison between our approach and more traditional matching methods, see the Discussion. We ensured that balance had been met between treatment and control territories for all covariates, with covariate means meeting a threshold of ±0.1. Since each treatment group was matched with its own unique set of private properties, effect sizes cannot be compared between models.

### Data

Land tenure regime data were sourced from the Atlas da Agropecuária Brasileira from Imaflora[50]. Imaflora has produced an extensive pre-processed dataset that integrated and standardized 18 diverse sources of land tenure regime data and systematically resolved geospatial overlaps. We merged several of the more specific Imaflora land regime classifications to represent the broader classifications of Indigenous lands, *Quilombola* territories, agrarian-reform settlements, private

properties, protected areas, and private properties (See SI). Indigenous lands and *Quilombola* territories included those with both formal and informal land tenure status, and protected areas included both strict protection and sustainable use. We included all territories from the land regime classifications in the Atlantic Forest listed above with a size larger or equal to 0.1 hectares.

The Imaflora territorial boundaries represented the most recent comprehensive data available, which was largely sourced from 2017 to 2018. This dataset did not include information on when land formalization occurred in time. Although some territories may have experienced privatization, changes in ownership, territory divisions, or formalization over time, we assumed that the broader land tenure regime (e.g., private property, protected area, Indigenous lands, QT or agrarian-reform settlement) did not transition over the study period. Other studies have made this same assumption[20]. Assessing transitions in land tenure regimes was outside the empirical scope of this study.

We sourced forest restoration data from Collection 8 of the Map-Biomas Project[51]. The MapBiomas forest data is comprised of 30 m resolution annual data of natural forest cover—including data on forest loss and forest gain—which includes pixels classified as dense, open, and mixed ombrophylous forest, semi-deciduous seasonal forest, deciduous seasonal forest, and pioneer tree formations, but does not include savannah, mangroves, and forest plantations. There is currently no way to detect the difference between active restoration and natural regeneration using satellite data. We defined long-term restoration gains as restoration (e.g., forest gain) that persisted until the end of the time period (2022) and that persisted for a minimum of 10 years. We defined restoration reversal as restoration that persisted between 3 and 10 years and was deforested before the end of the time period. We calculated long-term restoration gains and restoration reversals from pixels classified as natural forest cover across different years using an algorithm in Google Earth Engine. We did not include restoration that persisted for more than 10 years but was later deforested in our definition in order to solely isolate the unique effect of short-term reversals and long-term forest restoration. Since the vast majority of the restoration reversals only persisted between 4 and 8 years in the Atlantic Forest[37], our definition accounts for the restoration patterns that have most tended to occur in this biome. In addition, forests that are reversed after 10 years of age represent illegal deforestation according to the 2006 Atlantic Forest law, rather than a restoration reversal. Moreover, the definition of restoration reversals with a minimum of three years of age eliminates a bias towards restoration that occurred towards the end of the study period and did not have as much time to later be reversed. For both long-term restoration gains and reversals, pixels were classified as restoration if they were non-forested for at least three consecutive years before restoration occurred, and if this forest persisted for at least three additional years[37]. We excluded all restoration with fewer than 11 spatially connected pixels. This is the same approach used by similar studies[35–37]. Variables were aggregated to each land tenure regime in Google Earth Engine and were summed per territory to represent total area in 2022.

Model covariates included territory size, percentage of forest cover in 1985, temperature, precipitation, slope, elevation, distance to roads, distance to rivers, distance to nearest city, and population density[50,51,90–92]. Variables were selected based on data availability and precedence from prior studies[20,31,36,37,93]. Specifically, a list of possible covariates was compiled from a literature review of similar studies, and covariates were included in the analysis that were possible to collect through publicly-available data sources. See SI for information on covariates, data sources, and variable definitions (Table S2). Covariates were collected using Google Earth Engine, with the exception of territory size, which was sourced from Imaflora. All covariates were aggregated from the mean of all pixels in a territory (with the exception of territory size, which did not require aggregation). We excluded territories that did not have data for one or more of the control

variables, which represented 5.4% of territories in the original Imaflora dataset, and resulted in a final sample size of 1,887,057 territories to draw from for matching. Sample sizes by land tenure regime were: Indigenous lands-143, agrarian-reform settlements-1408, *Quilombola* territories-100, protected areas-697, and we drew from a sample of 1,884,269 private properties.

## Matching

We introduced a technique called agglomerative matching to estimate the average treatment effect on the treated (ATT) and to estimate trends specifically for treatment groups. Agglomerative matching is suitable for situations where typical matching techniques are unable to achieve adequate balance without dropping units due to the geographically small size of control units, and when the control units are relatively numerous and can be aggregated. Agglomerative matching is similar to synthetic control methods[94], regionalization[95], and knapsack problems[96]. However, unlike synthetic control, rather than blending the control units through averaging, we combine them through summing. From another direction, regionalization is a type of unsupervised learning that involves clustering geographically similar territories, such that the resulting clusters differ from one another[95]. Unlike regionalization, however, agglomerative matching requires matching the aggregated lands to a designated target land, a problem that is hard due to its resemblance to the subset sum and knapsack problems[96].

Indigenous lands, QT protected areas, and agrarian-reform settlements served as the four treatment conditions and private properties served as the control condition. We aimed to achieve covariate balance where the distributions of covariates in control and treatment conditions were approximately equal, without having to discard any treatment data. See SI for an explanation of other matching approaches attempted and considered, and why balance for other approaches was insufficient.

Agglomerative matching is a greedy algorithm that constructs an agglomerated control $C$ for each treatment land $t$. To create $C$, we iteratively combine control lands into $C$ based on how much their inclusion brings $C$ closer to $t$ in the L2-norm of the standardized covariates. Certain variables, such as area, are summed during agglomeration ("summable covariates"), and the remaining variables, such as average temperature, are updated through an area-weighted average ("non-summable covariates"). Due to the large number of control lands $N_C$ in the data (1,887,057), it is computationally infeasible to consider every single control land for inclusion into $C$ at each iteration, which would lead to an algorithm of runtime $O(N_T N_C^2)$, where $N_T$ is the number of treatment lands. Instead, we construct an approximate nearest neighbor index, so that we can query for the $K<<N$ control lands that are most similar to t in near-constant time, leading to an algorithm of runtime $O(N_T K^2)$. We used the *annoy* library developed by Spotify[97] to perform the approximate nearest neighbor computation, and for our analysis, we chose $K = 10,000$. We repeated this construction 500 times for each treatment land to quantify the stochastic error introduced by the approximate nearest neighbor search.

## Algorithm

Our algorithm for combining the covariates of control private properties proceeded according to the following steps:

1. Add control lands to an approximate nearest neighbors index using their non-summed covariates.
2. For each treatment land $t$:
   a. Using the approximate nearest neighbors index, create a candidate list of control lands, of size $K$, that are the most similar to $t$ in non-summed covariates.
   b. Find an approximate subset of the candidate list that matches $t$ in the following way:
      i. Create an agglomerated control $C$, which starts empty.

 ii. For each candidate control land *c*, sum its summable covariates with *C* and perform an area-weighted average of the non-summable covariates with *C* to create *C**.

 iii. Select the candidate control land *c* that brings the overall L2 distance between the covariates of *C** and *t*. Update *C* = *C**, and remove *c* from the candidate list.

 iv. Repeat, constructing *C* by greedily aggregating more control lands from the candidate list.

 v. Stop if the area of *C** is larger than the area of *t*, or if there is no *c* that can decrease the distance between *C* and *t*.

## Analysis

After constructing agglomerated controls, we directly compared the ATT on the matched pairs of treatment and control lands and reported the ATT upon re-matching using OFM (MatchIt package in R). We conducted OFM in the case that it produced better covariate balance between the treatment and agglomerated controls. OFM is a distance matching method that minimizes within-pair distances of covariates.

In both cases, we ran a linear regression of long-term restoration gains and restoration reversals on all covariates for all pairs of matched samples to estimate the treatment effect, performing a doubly-robust estimation[98]. We used the lm() function to fit the outcome and avg_comparisons() (marginaleffects package in R) to estimate ATT, using cluster-robust standard errors for OFM.

## Balance

Achieving covariate balance allows estimates to be less sensitive and closer to the true treatment effect[99]. We assessed balance for each individual covariate using SMDs, visual diagnostics, and prognostic scores. We recorded the maximum absolute SMD (the scaled and standardized differences in the means of each covariate between treatment groups) for all covariates in each seed. SMDs close to zero indicated good balance. Then, for each condition, we took the seed with the minimal maximum (mini-max) absolute SMD, yielding the agglomerative match that minimized the distances between treatment and control covariates. We reported the outcomes from the seed with the best covariate balance after matching. We took visual diagnostics using "Love plots", eCDF plots, eQQ plots, and kernel density plots (see SI). To ensure that the results were not idiosyncratic to one single run of agglomerative matching, we confirmed that the results were very similar to the average effect sizes from 500 runs of agglomerative matching, both with OFM and without OFM (maintaining the original one-to-one matches from the algorithm), and with or without discarding units outside of the region of common support (Table S2).

## Reporting summary

Further information on research design is available in the Nature Portfolio Reporting Summary linked to this article.

# Data availability

Our dataset was assembled from publicly and openly available sources.

# Code availability

Code to reproduce the model results is hosted at https://github.com/samzhang111/agglomerative-matching-reforestation. Zenodo archive with both code and data is available at https://doi.org/10.5281/zenodo.16516667.

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

## Acknowledgements

R.B. was supported by funding from a National Science Foundation (NSF) Social, Behavioral, and Economic Sciences (SBE) Postdoctoral Research Fellowship (SPRF) (NSF 18-584 Award #2203898). S.Z. was supported by an NSF Graduate Research Fellowship Award DGE-2040434 and by the US Research Software Sustainability Institute (URSSI) via grant G-2022-19347 from the Sloan Foundation. We thank Peter Newton for providing extensive feedback on paper drafts.

## Author contributions

R.B.: conceptualization, methodology, software, formal analysis, investigation, data curation, writing—original draft, writing—review and

editing, visualization, project administration, and funding acquisition. S.Z.: methodology, software, formal analysis, investigation, data curation, writing—original draft, writing—review and editing, and visualization. P.P.: methodology, software, data curation, writing—review and editing. M.M.N.: conceptualization, methodology, writing—review and editing, supervision.

## Competing interests

The authors declare no competing interests.
