## [Transparent Peer Review file · Nature Communications]

Land tenure regimes influenced long-term restoration gains and reversals across Brazil's Atlantic Forest

Corresponding Author: Dr Rayna Benzeev

Version 0:

Reviewer comments:

Reviewer #1

(Remarks to the Author)

1. Overview

This manuscript discusses the impact of different land tenure regimes in protecting Brazil's Atlantic Forest (Mata Atlantica in what follows). The authors compare the long-term restoration actions in indigenous lands and agrarian reform settlements with what happened in private lands.

This work relies on three key concepts: forest restoration, restoration successes, and restoration reversals. However, the text lacks a clear definition of forest restoration. There are implicit definitions of forest restoration, all of them slightly different:

- In lines 32-36, restoration is defined as any situation in which formerly deforested areas regrow back into a natural forest stage (either by abandonment or active recovery).
- In lines 41-44, the authors mention "restoration projects", presumably in situations where active recovery actions occur. However, it is unclear if they distinguish between active restoration and passive regrowth.
- In lines 54-66, the authors appear to distinguish between "active restoration" (line 59) and "natural regeneration" (line 61). Without defining them, they also mention "secondary forests" (line 61).

Thus, readers would benefit from a clear definition of the terms used in the term as early as possible in the manuscript. Authors should have answered the following questions in the Introduction and then use the resulting definition in their work:

- What is forest restoration in Mata Atlantica? When can an area of Mata Atlantica be considered that it has been restored? What is the criteria used (e.g., biomass, species diversity, biodiversity recovery)?
- What is the difference between a restoration project and a post-abandonment natural growth? Is this difference relevant to the research carried out by the authors? Can these two situations be distinguished?
- What is a secondary forest? Is a secondary forest a case of forest restoration? If so, what is the criteria used to distinguish them?

Another problem with the introduction is the lack of legal context in the manuscript. Since the research takes the land tenure regime as a basis for the analysis and conclusion, it would be appropriate for the authors to present the legal basis for land tenure in Brazil.

In the case of forest preservation and restoration, the authors need to present and discuss the role of Brazil's Forest Code (Law No. 12,651/2012). The code defines rules for forest preservation in private and public lands. For Mata Atlantica, private landowners are mandated to keep a minimum of 20% of their land as natural forests. The Law states that when landowners fail to meet this minimum requirement, they have to sign an agreement with the Public Attorney's Office where they pledge to restore their land.

In practice, implementation of the Forest Code is being delayed. Whatever the case, readers need to be informed of the legal

basis for forest restoration in Mata Atlantica. In addition, the authors also failed to provide additional context on the voluntary carbon offset markets which are starting to be implemented in Brazil. Such markets are considered a possible support for forest restoration in Mata Atlantica.

The authors claim to have “developed a new technique within the field of statistical matching” (line 109). However, they provide no supporting references to back their claim. Since the novelty of the method is not a key factor in the work's evaluation, that statement can be removed from the manuscript without a major loss of content. They should also compare their algorithms with similar proposals in the literature. Please see Aydin et al. (2021) for more details . The paper's results strongly rely on the spatial measurement of restoration success.

In the manuscript, the authors briefly describe how data was acquired: “We sourced forest restoration data from Collection 8 of the MapBiomas project” (line 407). This description is not enough to readers to judge the value of the work. MapBiomas does not provide data on forest restoration. Only maps of secondary vegetation are available. As outlined previously, the authors have conflated data on secondary vegetation as if all of this data were part of restoration projects, without explaining the difference to the reader.

Thus, the work must be revised to account for the difference. Using the term "restoration successes" suggests that landowners deliberately intended to increase forest areas in Mata Atlantica. However, the available data is insufficient to capture this intent.

2. Evaluation Questions

We now consider the questions posed by the journal to the reviewers.

a. What are the noteworthy results?

The authors present trends of forest restoration in the Brazilian Atlantic Forest and relate them to land tenure regimes. Based on their results, they make recommendations for public policy. However, they use data on the evolution of secondary vegetation provided by MapBiomas. This reviewer considers that the secondary vegetation data cannot be directly related to restoration projects. Therefore, the authors have made questionable decisions that weaken their results and conclusions.

b. Will the work be of significance to the field and related fields? How does it compare to the established literature? If the work is not original, please provide relevant references.

The work aims to provide novel insights that have not been published before in the literature.

c. Does the work support the conclusions and claims, or is additional evidence needed?

It does not seem to be so. The data used in the work is a map of secondary vegetation, which differs from a map of restoration projects. There is a mismatch between the original data and the author's interpretation. Thus, the conclusions are not supported by the data.

d. Are there any flaws in the data analysis, interpretation and conclusions? - Do these prohibit publication or require revision?

The flaw stems from the fact that the data available to the authors (yearly maps of secondary vegetation) does not support claims about restoration projects. There are various causes of an increase in secondary vegetation. Occasionally, land previously used for agriculture can be abandoned and left fallow. Secondary forests will develop in areas with specific climatic, soil, and topographic conditions. However, these situations differ from actual restoration projects, where landowners actively promote forest regrowth, sometimes motivated by the prospect of selling carbon credits.

The authors consider all cases of secondary vegetation to be associated with regeneration projects without justifying this choice. A primary flaw in this interpretation is the inconsistent public policy advice yielded by its results. The manuscript considers cases of secondary vegetation increase in indigenous lands, settlements, public properties and protected areas as equivalent for data analysis and uses them in regression analysis. Forest regrowth in indigenous lands will have different drivers than in protected areas.

The results produced by the regression model are hard to explain if considered only from the point of view of restoration successes and restoration reversal. One particular case concerns protected areas. The model finds that protected areas (PAs) are more associated with restoration reversals than successes. The authors explain: “One explanation for this trend is that PAs were already highly forested and may not have had much additional space for restoration to occur”. This argument illustrates the work's methodological problems. Is it relevant to consider that PAs are part of restoration projects in Mata Atlantica? To this reviewer, such approach leads to misleading public policy recommendations.

From the perspective of Brazil's Forest Code and also from Brazil's NDC, forest restoration is taken to be the actions that landowners must undertake to comply with the law. All private lands above a specific minimum size in Mata Atlantica must preserve 20% of their areas. By making private landowners comply with the law, Brazil expects to reach 12 Mha of forest restoration by 2030. From the public policy perspective, the authors must show how Brazil can best reach its stated goals. The work fails to do so in its current version.

f. Is the methodology sound? Does the work meet the expected standards in your field?

As stated above, the methodology is unsound insofar as it conflates all instances of secondary vegetation with forest restoration actions. This is a particular interpretation that fails to consider how Brazilian legislation (e.g., the Forest Code) and Brazil's commitments to NDC use the concept of forest restoration.

g. Is there enough detail provided in the methods for the work to be reproduced?

Yes. I reviewed the code.

3. Final Assessment

Because of the reasons stated above, the recommendation is for the paper to be rejected.

(Remarks on code availability)

Reviewer #2

(Remarks to the Author)

This manuscript aims to identify the impact of different land tenure regimes on forest restoration successes and reversals in the Brazilian Atlantic forest by using a novel statistical matching technique – agglomerative matching. The study finds that Indigenous lands typically had a much larger area of forest that was restored successfully than seen in private properties, while agrarian-reform settlements also showed similar increases. However, restoration reversals were also more pronounced under these land use tenures. Neither Quilombola territories nor protected areas showed statistically significant differences from private properties regarding restoration successes or restoration reversals. These results have the potential to support forest restoration by people that are typically marginalised by funded restoration projects.

I really enjoyed reading the manuscript. It is a great, applied idea that seems to be well implemented, and very well explained. However, I have a number of suggestions and doubts that the authors should aim to address.

Major issues

- Lines 128-161: Although the results are interesting, they are very short and this left me wanting more. Would it be possible to try and explain the variability in size of restoration successes and restoration reversals for each land tenure regime? For example, is there any information on management of the lands that could be used? Or could size or protected area perimeter be used? I'm not totally clear about whether this would work with agglomerative matching, but this comment was inspired by studies I've seen that used matching techniques to investigate protected area effectiveness and subsequently explored reasons for variability^{1,2}. This could potentially be used to address some of the hypotheses the authors set out in lines 190 – 210.
- As the authors acknowledge (lines 335-338), the sample sizes for indigenous lands and Quilombola territories are small when compared to other land use tenures. The authors state that this is because of missing data for these tenure systems. However, depending on what types of data are missing, it may be possible to overcome some of this problem by using multiple imputation methods^{3,4}. Without using imputation there is a danger that the authors' analysis uses a biased subset of sites that is not representative of these tenure systems as a whole⁵.
- Table S1 – I would find it really helpful to have information on how these different covariates vary by land use tenure. This would help to put the results of the study in context. For example, on lines 298-299 the authors argue that protected areas may not have contributed much in terms of restoration success because they may have already been highly forested, leaving little space for forest restoration. Surely this could be confirmed by examining the covariates, couldn't it?

Minor issues

- Table 1 – The information in this table is vital to understand the context of the work, but is there a way to display the information in a figure rather than a table?
- Table 2 – Similar to my comment above, could this table be turned into a figure? You could potentially take the information from the table and produce two separate plots for restoration successes and restoration reversals. I feel like this would help to communicate the message of the study much more effectively.
- Lines 164 – 363: I recommend not using many acronyms in the discussion as it makes it difficult to follow what is being said.
- Lines 423 – 425: I would appreciate more detail about why the covariates selected were used.
- Lines 435 – 502: There is no mention of the programs used for analyses aside from Google Earth Engine.

References

1. Wauchope, H. S. et al. Protected areas have a mixed impact on waterbirds, but management helps. *Nature* 605, 103–107 (2022).
2. Geldmann, J., Manica, A., Burgess, N. D., Coad, L. & Balmford, A. A global-level assessment of the effectiveness of protected areas at resisting anthropogenic pressures. *Proc. Natl. Acad. Sci. U. S. A.* 116, 23209–23215 (2019).

3. Oldekop, J. A., Holmes, G., Harris, W. E. & Evans, K. L. A global assessment of the social and conservation outcomes of protected areas: Social and Conservation Impacts of Protected Areas. *Conserv. Biol.* 30, 133–141 (2016).
4. Hajjar, R. et al. A global analysis of the social and environmental outcomes of community forests. *Nat. Sustain.* 4, 216–224 (2020).
5. Nakagawa, S. & Freckleton, R. P. Missing inaction: the dangers of ignoring missing data. *Trends Ecol. Evol.* 23, 592–596 (2008).

(Remarks on code availability)

Reviewer #3

(Remarks to the Author)

Key results

The paper demonstrates that Indigenous lands (ILs), and agrarian reform settlements (ARSs) had more long-term restoration successes than Private Properties (PPs) when controlling for biophysical conditions. ILs and ARSs concurrently had the higher rates of restoration reversal compared to PPs. Combined these results indicate that avoiding land tenure transition to PPs could improve the efficiency of long-term restoration outcomes.

From a methodological point of view the paper presents a new and innovative technique of agglomerative matching that iteratively combines multiple PPs into private agglomerates to achieve covariate balance when matching treatment and control territories. This enables to avoid dropping any data retaining the analysis of 1.9 million territories in The Atlantic Forest (AF) biome that is a top priority ecosystem for restoration.

Validity

Data are gathered from a representative sample (1.9 million territories) and are described as the most reliable in terms of what is currently available (e.g. Imaflora, Table S1). Methods are described in a transparent and reproducible way with limitations highlighted. Each model was run 500 times to compare standardized mean differences (SMD) and prognostic scores across iterations.

Significance

The implications of land tenure regimes for restoration are explained in context. The broader applications of the results for restoration decision-makers and practitioners are presented in a concise and effective way.

Data and methodology, and analytical approach

The validity of the approach looks sound, and the methodology is presented in an easy-to-follow way. I can understand the matching approach, the algorithm, analysis and balance. I also assessed the data provided in the supplementary material and they look robust. However, I am not an expert in matching techniques. The code needs to be checked by someone more expert (e.g. Table S2, Figure S1 and S2).

Suggested improvements, clarity and context

Abstract

In general, this could be made more effective by removing the reference to 189 ha and 21 ha. At first one doesn't understand why only a value for ILs is reported and not two (ILs and ARSs) since both are mentioned. So, my suggestion would be to just remove the values or alternatively provide both, but this could make the abstract too long.

Also, I would suggest to stress that ILs and ARSs concurrently had the higher rates of success and restoration reversal compared to PPs. How it is phrased at the moment, it almost looks like a contradiction. So, I would add 'concurrently or in combination' to reassure the reader that this is indeed not a contradiction. Specify that the total area of restoration successes was around ten times more than reversals.

Perhaps briefly explain the reasons too of this concurrent results (success and restoration reversals on the same type of land tenure).

The last sentence is too broad. I would suggest writing something more specific along the lines of ... Combined these results indicate that avoiding land tenure transition to PPs could improve the efficiency of long-term restoration outcomes.

Lines 14-20 to be rephrased in particular

[... We find that ILs and ARSs have significantly more long-term restoration successes than PPs (189 ha more per IL than PPs). Restoration reversals are also significantly higher in ILs and ARSs, although by a smaller margin (21 ha more restoration reversals per IL than PPs). By characterizing how restoration reversals and successes vary between land tenure regimes, our study may support efforts to improve the longevity of forest restoration].

Introduction

Lines 32-36 Can this be rephrased in a clearer way? As it is further down in the paper?

When evaluating forest restoration (hereafter restoration), one critical distinction is the difference between restoration successes—restored forests that remain forested for the long term (defined here as standing forests that have persisted for a minimum of 10 years and continued to persist), and restoration reversals—restored forests that are later deforested (defined here as persisting between 3-10 years—see Methods).

Here is much clearer ... We defined restoration success as restoration that persisted until the end of the time period (2022), and that persisted for a minimum of ten years. We defined restoration reversal as restoration that persisted between 3-10 years and was deforested before the end of the time period

Results

Lines 155-161 Descriptive statistics of restoration reversals and successes

I would move this at the beginning of the results

References

The results are provided with sufficient context and consideration of previous work. References to previous literature are presented appropriately.

(Remarks on code availability)

Version 1:

Reviewer comments:

Reviewer #1

(Remarks to the Author)

1. What are the noteworthy results?

The authors have significantly improved the manuscript and addressed all the points raised by the reviewers. The work now contributes to the understanding of land use changes in the Brazilian Mata Atlantica. The main result is the difference between restoration gains in indigenous lands compared with other tenure regimes.

2. Are there any flaws in the data analysis, interpretation and conclusions?

The revised version improved the data analysis, and led to conclusions which are tenable. The only concern is the use of the term "restoration success", which implies a deliberate action by the tenant to achieve a goal. As the authors note, they make no distinction between areas which have resulted from a restoration project and areas which have been abandoned. For this reason, the authors are requested to use a more neutral term, such as "restoration gain", instead of "restoration success".

3. Is the methodology sound? Does the work meet the expected standards in your field?

The methods appear to be sound, based on the results. However, the "Methods" section still needs a revision. The sentence "Our agglomerative matching algorithm iteratively sorts through private properties (see algorithm below) to select a combination of private properties with covariates similar to each treatment territory" introduces the concept of "treatment territory" that is neither described in the main text nor in SI. Please explain this term and use a more English-sounding concept. Would "spatial cluster" be more adequate. After all, what the authors are doing in to join similar PPs in a spatial cluster.

4. Is there enough detail provided in the methods for the work to be reproduced?

Yes, there is enough detail to reproduce the paper.

Overall recommendation: ACCEPT WITH MINOR REVIEWS

(Remarks on code availability)

The code is clear and easy for an R expert to understand.

Reviewer #2

(Remarks to the Author)

I am satisfied that the authors have addressed all the points that I raised in the previous round of review and that the manuscript is more robust as a result. Good work!

After reading the manuscript I found no issues with the current version.

(Remarks on code availability)

Reviewer #3

(Remarks to the Author)

I have reviewed the revised version of the manuscript and find that the authors have addressed all of my previous comments comprehensively and to a high standard. The revisions have improved the clarity and rigor of the work, and I am satisfied with the responses provided.

While I have not reviewed the accompanying code, based on the manuscript and the authors' detailed responses, I have no

further concerns, and I support the publication.
Best regards

(Remarks on code availability)

RESPONSE TO REVIEWERS

We thank the three reviewers for their careful consideration of our manuscript. We are grateful for their comments and have now incorporated their suggestions. **Our responses to each comment are shown in bold below. Line numbers refer to the revised manuscript.**

Reviewer #1 (Remarks to the Author):

1. Overview

This manuscript discusses the impact of different land tenure regimes in protecting Brazil's Atlantic Forest (Mata Atlantica in what follows). The authors compare the long-term restoration actions in indigenous lands and agrarian reform settlements with what happened in private lands.

This work relies on three key concepts: forest restoration, restoration successes, and restoration reversals. However, the text lacks a clear definition of forest restoration. There are implicit definitions of forest restoration, all of them slightly different:

- In lines 32-36, restoration is defined as any situation in which formerly deforested areas regrow back into a natural forest stage (either by abandonment or active recovery).
- In lines 41-44, the authors mention “restoration projects”, presumably in situations where active recovery actions occur. However, it is unclear if they distinguish between active restoration and passive regrowth.
- In lines 54-66, the authors appear to distinguish between “active restoration” (line 59) and “natural regeneration” (line 61). Without defining them, they also mention “secondary forests” (line 61).

Thus, readers would benefit from a clear definition of the terms used in the term as early as possible in the manuscript.

Thank you for your review and for your suggestion to provide a clearer definition of forest restoration. We have now added a clear definition into the first paragraph of the Introduction. Lines 32-33: “defined as forest gain, including any change from productive land uses to natural forest cover⁷”.

We additionally have now reinforced this definition by adding that forest gain included both active restoration and natural regeneration at the end of the Introduction.

Line 132: “measuring restoration that occurred through both active restoration and natural regeneration”.

We have now added a definition of restoration projects.

Lines 40-41: “e.g., planned initiatives to intentionally plant and restore forests”.

We have now added an additional sentence to clarify that forest restoration has been defined as both active restoration and natural regeneration in past studies, and that natural regeneration can occur either intentionally or unintentionally.

Lines 64-65: “Importantly, both active restoration and natural regeneration have been considered forest restoration (according to our definition), since both have resulted in forest gain^{7,35-38}.”

We additionally added the phrase “either intentionally or unintentionally” to the definition of natural regeneration to indicate that natural regeneration does not only occur through land abandonment, but also when people allow forests to regrow naturally.

Lines 59-61: “much restoration has occurred through natural regeneration²⁹⁻³²—a process where degraded and abandoned agricultural land regenerates naturally into forest either intentionally or unintentionally³³.”

Last, we deleted the term ‘secondary forest’, replacing this term with the phrase ‘a large percentage of forests’, as this was more specific and we had previously only used the term one time in our manuscript. The sentence now reads:

Line 63: “particularly in locations where a large percentage of forests remain standing^{29-31,34}.”

Authors should have answered the following questions in the Introduction and then use the resulting definition in their work:

- What is forest restoration in Mata Atlantica? When can an area of Mata Atlantica be considered that it has been restored? What is the criteria used (e.g., biomass, species diversity, biodiversity recovery)?

We have now added an additional sentence into the Introduction which grounds our definition of restoration (areas of forest gain) in the literature of restoration in the Atlantic Forest.

Lines 64-65: Importantly, both active restoration and natural regeneration have been considered forest restoration (according to our definition), since both have resulted in significant forest gain^{7,35-40}.

We also explain that other studies measuring forest restoration in the Atlantic Forest using satellite imagery have used measures of restoration as a determinant of restored ecosystem services and biodiversity recovery (Rosa et al., 2021).

Lines 77-78: “Large-scale analyses of restoration in the Atlantic Forest have used measures of forest gain as a determinant of restored ecosystem services and biodiversity recovery^{31,35}”

We additionally now cite the specific definition of forest restoration in the Atlantic Forest from Crouzeilles et al., 2019, which corresponds directly to our definition of forest restoration that we have now added to the first paragraph of the Introduction. If you would like us to make our definition longer and more specific (like the Crouzeilles definition below) then we would be happy to do so.

The definition from Crouzeilles et al., 2019 is:

“To estimate the total amount of “restored forest” (defined here as native forests under either passive or active recovery that may not have yet reached the attributes of reference ecosystems) in the Brazilian Atlantic Forest between 2011 and 2017, we used a 30 m resolution land use and cover mapping product (“Mapbiomas”, version 3.0; http://mapbiomas.org/pages/database/mapbiomas_collection). Mapbiomas is a collaborative initiative involving Non-Governmental Organizations, private companies, and research organizations to annually monitor land use change across all Brazilian biogeographical regions. Five conservative criteria were used to identify restored forest. The area should: (i) be previously classified as agriculture or pasture for at least five consecutive years; (ii) have at least five connected pixels also classified as restored forest; (iii) be classified as forest in 2017; (iv) be classified as forest for at least three consecutive years; and (v) should not be a commercial tree plantation”

• What is the difference between a restoration project and a post-abandonment natural growth? Is this difference relevant to the research carried out by the authors? Can these two situations be distinguished?

We have now added more detail to clarify that we included both restoration projects and post-abandonment natural growth (what we call ‘natural regeneration’) in our definition of restoration. Our choice to analyze both active restoration and natural regeneration in combination is supported by many other studies that have analyzed all cases of forest gain using MapBiomias data, including both active restoration and natural regeneration in their analyses (Crouzeilles et al., 2019; Rosa et al., 2021; Piffer et al., 2021; Piffer et al., 2022; Benzeev et al., 2023). Since both active restoration and natural regeneration appear identical in satellite imagery, all studies that have relied on MapBiomias data to quantify forest gain do not distinguish between these two types of restoration.

Lines 32-33 “defined as forest gain, including any change from productive land uses to natural forest cover⁷”

Lines 64-65: “Importantly, both active restoration and natural regeneration have been considered forest restoration (according to our definition), since both have resulted in forest gain^{7,35-38}.”

Lines 77-78: “Large-scale analyses of restoration in the Atlantic Forest have used measures of forest gain as a determinant of restored ecosystem services and biodiversity recovery^{31,35}.”

Line 132: “measuring restoration that occurred through both active restoration and natural regeneration”

Lines 463-464: “There is currently no way to detect the difference between active restoration and natural regeneration using satellite data.”

As we have now added to the paper, there is currently no way to detect the difference between active restoration and natural regeneration using satellite data. To our knowledge, only one paper analyzing restoration using geospatial data has distinguished between active restoration and natural regeneration, and they did so by overlaying the MapBiomias dataset with additional maps of commercial forestry and initiatives registered with the ‘Atlantic Forest Restoration Pact’ from other data sources (Crouzeilles et al., 2020). Although this approach was able to separate restoration initiatives registered with ‘the Pact’ from natural regeneration, it could not separate the many active restoration initiatives that were not registered in the Pact from natural regeneration.

In our study, we chose to analyze all areas of forest gain, as this had more precedence from past studies (Crouzeilles et al., 2019; Rosa et al., 2021; Piffer et al., 2021; Piffer et al., 2022; Benzeev et al., 2023; Bicuda da Silva et al., 2023; Camara et al., 2023).

In addition, we do not believe that distinguishing between these two types of restoration is relevant to our study. Our research question asks whether forest gain occurs and persists, and does not ask about the mechanisms by which it does so. Given that there is a spectrum by which active restoration and natural regeneration occur (since natural regeneration can occur through ‘assisted natural regeneration’—where people intentionally assist forests to naturally regenerate), we believe that this distinction is not extremely relevant, and the most reasonable measurements to make are whether or not forests are restored and are able to persist. However, we had previously included a detailed section in the Discussion to discuss the possible mechanisms of our results (lines 229-252).

• What is a secondary forest? Is a secondary forest a case of forest restoration? If so, what is the criteria used to distinguish them?

We no longer use the term secondary forest in our manuscript. Previously, the term only appeared one time. We believe that distinguishing between restoration reversals and successes is sufficient to separate long-term and short-term forest regrowth. We additionally now specify that the MapBiomass data we utilized is able to directly measure forest gain over time (rather than only mapping secondary forest cover).

Lines 460-464: “The MapBiomass forest data is comprised of 30-m resolution annual data of natural forest cover—including data on forest loss and forest gain—which includes pixels classified as dense, open, and mixed ombrophylous forest, semi-deciduous seasonal forest, deciduous seasonal forest, and pioneer tree formations, but does not include savannah, mangroves, and forest plantations. There is currently no way to detect the difference between active restoration and natural regeneration using satellite data.”

See Rosa et al., 2021: “[Mapbiomas] also allows for a long-term, in-depth analysis of yearly forest dynamics over large spatial scales, not only improving our understanding of forest spatial structure in tropical regions but also giving support for a better quantification of restoration benefits”

Another problem with the introduction is the lack of legal context in the manuscript. Since the research takes the land tenure regime as a basis for the analysis and conclusion, it would be appropriate for the authors to present the legal basis for land tenure in Brazil.

In the case of forest preservation and restoration, the authors need to present and discuss the role of Brazil’s Forest Code (Law No. 12,651/2012). The code defines rules for forest preservation in private and public lands. For Mata Atlantica, private landowners are mandated to keep a minimum of 20% of their land as natural forests. The Law states that when landowners fail to meet this minimum requirement, they have to sign an agreement with the Public Attorney’s Office where they pledge to restore their land.

In practice, implementation of the Forest Code is being delayed. Whatever the case, readers need to be informed of the legal basis for forest restoration in Mata Atlantica. In addition, the authors also failed to

provide additional context on the voluntary carbon offset markets which are starting to be implemented in Brazil. Such markets are considered a possible support for forest restoration in Mata Atlantica.

We have now added four sentences to the Introduction to introduce the Forest Code and legal context (including voluntary carbon markets) into the manuscript.

Lines 113-125: “In addition, most programs and policies on restoration in Brazil influence private properties and most planned restoration initiatives in Brazil have taken place in private properties. For example, Brazil’s Forest Code mandates that all private properties in the Atlantic Forest maintain 20% of their property forested and for noncompliant landowners to conduct restoration. Other programs such as the Atlantic Forest law, increased monitoring and enforcement, the Atlantic Forest Restoration Pact, voluntary carbon offset markets, and other initiatives aimed at achieving Brazil’s Nationally Determined Contribution to mitigate climate change have also principally targeted restoration of private properties. Although private properties have the largest restoration debt due to their significant contribution to deforestation, and some programs should not be applied to other land tenure regimes due to their social consequences^{14,51,52}, only focusing restoration initiatives in private properties prevents restoration financing from reaching these other land tenure regimes.”

The authors claim to have “developed a new technique within the field of statistical matching” (line 109). However, they provide no supporting references to back their claim. Since the novelty of the method is not a key factor in the work's evaluation, that statement can be removed from the manuscript without a major loss of content. They should also compared their algorithms with similar proposals in the literature. Please see Aydin et al. (2021) for more details .

The paper’s results strongly rely on the spatial measurement of restoration success.

We have now deleted the word “new” from the phrase you suggested. We have also now added a few sentences that reference algorithms from similar papers in the literature, including the regionalization literature you suggested (Aydin et al., 2021).

Lines 503-509: “Agglomerative matching is similar to synthetic control methods⁹¹, regionalization⁹², and knapsack problems⁹³. However, unlike synthetic control, rather than blending the control units through averaging, we combine them through summing. From another direction, regionalization is a type of unsupervised learning that involves clustering geographically similar territories, such that the resulting clusters differ from one another⁹². Unlike regionalization, however, agglomerative matching requires matching the aggregated lands to a designated target land, a problem that is hard due to its resemblance to the subset sum and knapsack problems⁹³.”

In the manuscript, the authors briefly describe how data was acquired: “We sourced forest restoration data from Collection 8 of the MapBiomas project” (line 407). This description is not enough to readers to judge the value of the work. MapBiomas does not provide data on forest restoration. Only maps of secondary vegetation are available. As outlined previously, the authors have conflated data on secondary vegetation as if all of this data were part of restoration projects, without explaining the difference to the reader.

The MapBiomass dataset includes annual maps of secondary vegetation that enable us to collect yearly measurements of forest loss and forest gain. The MapBiomass annual secondary vegetation measurements follow a similar reasoning as that of our study and the same definitions of the previous studies that also used the MapBiomass dataset to measure restoration defined as forest gain (Crouzeilles et al., 2019; Crouzeilles et al., 2020; Piffer et al., 2021; Rosa et al., 2021; Piffer et al., 2022; Benzeev et al., 2023; Camara et al., 2023; Bicuda da Silva et al., 2023). This approach has been widely adopted by the scientific literature and has been developed in collaboration with the technical staff from the MapBiomass project. We have now added an additional sentence that thoroughly describes the data sourced from MapBiomass and how this data directly measures restoration according to our definition of restoration (forest gain that has occurred through both active restoration and natural regeneration).

Lines 460-464: “The MapBiomass forest data is comprised of 30-m resolution annual data of natural forest cover—including data on forest loss and forest gain—which includes pixels classified as dense, open, and mixed ombrophylous forest, semi-deciduous seasonal forest, deciduous seasonal forest, and pioneer tree formations, but does not include savannah, mangroves, and forest plantations. There is currently no way to detect the difference between active restoration and natural regeneration using satellite data.”

Lines 467-469: “We calculated *restoration successes* and *restoration reversals* from pixels classified as natural forest cover across different years using an algorithm in Google Earth Engine.”

We had also previously described how we calculated forest restoration (e.g., forest gain) in a more robust way, in order to only measure forest restoration that remained forested for at least three years, and that occurred in a patch as large or larger than 11 pixels.

Lines 477-480: “For both restoration reversals and successes, pixels were classified as restoration if they were non-forested for at least three consecutive years before restoration occurred, and if this forest persisted for at least three additional years³⁵. We excluded all restoration with fewer than 11 spatially connected pixels. This is the same approach used by similar studies³⁵⁻³⁷”

Thus, the work must be revised to account for the difference. Using the term “restoration successes” suggests that landowners deliberately intended to increase forest areas in Mata Atlantica. However, the available data is insufficient to capture this intent.

The definition we previously included for the term “restoration successes” is “restoration that persisted for a minimum of ten years and until the end of the time period (2022).” This definition does not imply that landowners deliberately intended to increase forest areas. Instead, it implies that restoration has successfully occurred regardless of the intention of landowners. If you and the Editor believe that we need to change this term then we would be happy to do so.

2. Evaluation Questions

We now consider the questions posed by the journal to the reviewers.

a. What are the noteworthy results?

The authors present trends of forest restoration in the Brazilian Atlantic Forest and relate them to land tenure regimes. Based on their results, they make recommendations for public policy. However, they use data on the evolution of secondary vegetation provided by MapBiomias. This reviewer considers that the secondary vegetation data cannot be directly related to restoration projects. Therefore, the authors have made questionable decisions that weaken their results and conclusions.

We have now added additional detail into the Methods section in order to clarify that MapBiomias data measures restoration (defined as forest gain) and that this decision has been supported by many past studies who have also used these same measurements. Please refer to our explanations above.

b. Will the work be of significance to the field and related fields? How does it compare to the established literature? If the work is not original, please provide relevant references.

The work aims to provide novel insights that have not been published before in the literature.

c. Does the work support the conclusions and claims, or is additional evidence needed?

It does not seem to be so. The data used in the work is a map of secondary vegetation, which differs from a map of restoration projects. There is a mismatch between the original data and the author's interpretation. Thus, the conclusions are not supported by the data.

Our definition of restoration does not aim to only measure active restoration projects, but all forest gain, which includes both active restoration and natural regeneration. In addition, it is possible to directly measure forest gain using the MapBiomias dataset, rather than only analyzing secondary vegetation cover. Please see our detailed explanations above.

d. Are there any flaws in the data analysis, interpretation and conclusions? - Do these prohibit publication or require revision?

The flaw stems from the fact that the data available to the authors (yearly maps of secondary vegetation) does not support claims about restoration projects. There are various causes of an increase in secondary vegetation. Occasionally, land previously used for agriculture can be abandoned and left fallow. Secondary forests will develop in areas with specific climatic, soil, and topographic conditions. However, these situations differ from actual restoration projects, where landowners actively promote forest regrowth, sometimes motivated by the prospect of selling carbon credits.

Again, please see our detailed responses to these points above. We now clearly state that forest restoration includes any gains in forest cover being through either passive natural regeneration or active restoration efforts.

The authors consider all cases of secondary vegetation to be associated with regeneration projects without justifying this choice. A primary flaw in this interpretation is the inconsistent public policy advice yielded by its results. The manuscript considers cases of secondary vegetation increase in indigenous lands,

settlements, public properties and protected areas as equivalent for data analysis and uses them in regression analysis. Forest regrowth in indigenous lands will have different drivers than in protected areas.

In terms of clarifying the link between the results of our analysis and implications for policy, we had previously included several broader applications for restoration decision-making and practice that specifically aim to inform practice to facilitate natural regeneration, rather than only aiming to inform active restoration projects. See below:

Lines 370-374: “It is therefore important to a) protect existing communal land tenure regimes from threats of land grabbing and privatization, b) more explicitly promote conditions for natural regeneration (rather than a sole focus on active planting) in communal land tenure regimes, and c) reconsider strengthened land and self-determination rights for communal territories as a distinct restoration strategy.”

In terms of the drivers of forest restoration between land tenure regimes, our study focused on analyzing the differences between patterns of restoration between land tenure regimes, and not on identifying the different drivers of restoration between land tenure regimes. We had previously described some of these possible drivers and mechanisms in great detail in the Discussion (lines 218-240). We agree that drivers likely differ between land tenure regimes, and we observe the differences in these drivers by measuring differences in restoration outcomes (restoration reversals and successes). Our modifications to the manuscript explicitly set the expectation that our goal was to identify restoration patterns and not restoration drivers.

The results produced by the regression model are hard to explain if considered only from the point of view of restoration successes and restoration reversal. One particular case concerns protected areas. The model finds that protected areas (PAs) are more associated with restoration reversals than successes. The authors explain: “One explanation for this trend is that PAs were already highly forested and may not have had much additional space for restoration to occur”. This argument illustrates the work’s methodological problems. Is it relevant to consider that PAs are part of restoration projects in Mata Atlantica? To this reviewer, such approach leads to misleading public policy recommendations.

We have now added an additional figure in the SI that clarifies patterns of forest cover in PAs in our study (Fig. S1).

In addition, as mentioned above, we are not only evaluating ‘restoration projects’ in the Atlantic Forest, but also forest gains that occurred in the Atlantic Forest, either by natural regeneration (passive) or active restoration. As such, it makes sense to compare restoration (as measured by forest gain) between land tenure regimes, including PAs. We agree with you that PAs have not been a part of formal restoration projects. The other land tenure regimes included in our study (Indigenous lands, agrarian-reform settlements, and *Quilombola* territories) have also not—for the most part—participated in formal restoration projects, which have principally taken place in private properties. This is why our study aimed to analyze natural regeneration in addition to restoration projects.

In terms of the public policy recommendations, it may not make sense to rely upon high amounts of future restoration to occur in PAs since they are already so highly forested. Other lands that have lower percentages of forest cover may experience higher rates of future restoration. We previously highlighted this point in our Discussion (lines 335-347).

From the perspective of Brazil's Forest Code and also from Brazil's NDC, forest restoration is taken to be the actions that landowners must undertake to comply with the law. All private lands above a specific minimum size in Mata Atlantica must preserve 20% of their areas. By making private landowners comply with the law, Brazil expects to reach 12 Mha of forest restoration by 2030. From the public policy perspective, the authors must show how Brazil can best reach its stated goals. The work fails to do so in its current version.

We have now added an additional sentence to speak to how Brazil can best reach its goals of 12 Mha of forest restoration by 2030.

Lines 376-379: "Although initiatives in private properties will be essential to achieving Brazil's international restoration commitments and to compensate for high rates of past deforestation, restoration that has occurred and persisted on other land tenure regimes have also contributed towards achieving these commitments."

f. Is the methodology sound? Does the work meet the expected standards in your field?

As stated above, the methodology is unsound insofar as it conflates all instances of secondary vegetation with forest restoration actions. This is a particular interpretation that fails to consider how Brazilian legislation (e.g., the Forest Code) and Brazil's commitments to NDC use the concept of forest restoration.

Please see our explanation above regarding our definition of forest restoration.

We also now reference the Forest Code and the NDC in the Introduction.

Lines 113-125: "In addition, most programs and policies on restoration in Brazil influence private properties and most planned restoration initiatives in Brazil have taken place in private properties. For example, Brazil's Forest Code mandates that all private properties in the Atlantic Forest maintain 20% of their property forested and for noncompliant landowners to conduct restoration. Other programs such as the Atlantic Forest law, increased monitoring and enforcement, the Atlantic Forest Restoration Pact, voluntary carbon offset markets, and other initiatives aimed at achieving Brazil's Nationally Determined Contribution to mitigate climate change have also principally targeted restoration of private properties. Although private properties have the largest restoration debt due to their significant contribution to deforestation, and some programs should not be applied to other land tenure regimes due to their social consequences^{14,49,50}, only focusing restoration initiatives in private properties prevents restoration financing from reaching these other land tenure regimes."

While Brazil's Forest Code and commitments to NDC specifically target restoration occurring in private properties, our paper highlights the fact that restoration does not only occur in private

properties. In fact, long-term restoration successes have occurred at higher rates in Indigenous lands and agrarian-reform settlements.

g. Is there enough detail provided in the methods for the work to be reproduced?

Yes. I reviewed the code.

We are glad to hear that you believe our methods are reproducible.

3. Final Assessment

Because of the reasons stated above, the recommendation is for the paper to be rejected.

We have now addressed all of your main concerns and hope you will reevaluate your initial assessment of our manuscript.

Reviewer #2 (Remarks to the Author):

This manuscript aims to identify the impact of different land tenure regimes on forest restoration successes and reversals in the Brazilian Atlantic forest by using a novel statistical matching technique – agglomerative matching. The study finds that Indigenous lands typically had a much larger area of forest that was restored successfully than seen in private properties, while agrarian-reform settlements also showed similar increases. However, restoration reversals were also more pronounced under these land use tenures. Neither Quilombola territories nor protected areas showed statistically significant differences from private properties regarding restoration successes or restoration reversals. These results have the potential to support forest restoration by people that are typically marginalised by funded restoration projects.

I really enjoyed reading the manuscript. It is a great, applied idea that seems to be well implemented, and very well explained. However, I have a number of suggestions and doubts that the authors should aim to address.

Thank you very much for your review. We are pleased to hear that you really enjoyed reading the manuscript and believe that it is a great applied idea that has been well implemented and well explained.

Major issues

- Lines 128-161: Although the results are interesting, they are very short and this left me wanting more. Would it be possible to try and explain the variability in size of restoration successes and restoration reversals for each land tenure regime? For example, is there any information on management of the lands that could be used? Or could size or protected area perimeter be used? I'm not totally clear about whether this would work with agglomerative matching, but this comment was inspired by studies I've seen that

used matching techniques to investigate protected area effectiveness and subsequently explored reasons for variability^{1,2}. This could potentially be used to address some of the hypotheses the authors set out in lines 190 – 210.

We have now included an additional figure in the SI that explores variability in terms of each of the covariates (Fig. S1). We have also now included an additional sentence that refers to this figure, mentions variability, and refers to the SI.

Lines 159-160: “See SI to visualize the variability of model covariates (Fig. S1).”

In terms of your suggestion to analyze the variable territory size, we already had included this variable in our model as a covariate. We saw that the two papers you suggested also analyzed this variable, and thus our analysis is consistent with these two papers you recommended.

In terms of analyzing a variable representing management of lands, there is currently no publicly-available data on land management for all land tenure regimes. Once more data becomes available on land management by land tenure regime, we agree that it would be interesting to analyze trends in restoration reversals and successes according to land management strategies.

In terms of analyzing a variable representing area perimeter, to our knowledge no other similar study has included this variable as a covariate. Moreover, the two studies you suggested (Wauchope et al., 2022; Geldmann et al., 2019) also did not include this variable, which suggests that excluding this variable is supported by the literature. We had previously mentioned the idea of including this variable in our Discussion section as a potential area of future research.

If you and the Editor believe that our Results section is too short, we would be happy to move some of the content that is currently in the SI into the Results. Given the word count restrictions of *Nature Communications*, we had hoped to summarize our Results in as few words as possible in order to have more words to describe the possible mechanisms behind these results in the Discussion.

- As the authors acknowledge (lines 335-338), the sample sizes for indigenous lands and Quilombola territories are small when compared to other land use tenures. The authors state that this is because of missing data for these tenure systems. However, depending on what types of data are missing, it may be possible to overcome some of this problem by using multiple imputation methods^{3,4}. Without using imputation there is a danger that the authors’ analysis uses a biased subset of sites that is not representative of these tenure systems as a whole⁵.

The lack of available data on *Quilombola* territories (QTs) is important, however all past studies that have analyzed *Quilombola* territories have also faced this same limitation (e.g., Pacheco & Meyer, 2022). For most of the 5000+ QTs across Brazil, there aren’t even delimited spatially-explicit polygons showing where these territories are located. As such, multiple imputation would not be possible. These data are missing because a territory is not considered to be a QT unless they have started the process of obtaining tenure, and each of the 24 states containing QTs in Brazil have different protocols and priorities for tenure. As such, the tenure process differs greatly by state,

even though the process is federally administered. Some states (e.g., Pará, Maranhão, and Bahia), have granted incomplete tenure for many QTs, while other states (e.g., Santa Catarina and Rondônia) have granted very few or none.

We had previously highlighted the constraints of this missing data in the Discussion:

Lines 327-334: “However, there is very little data available on *Quilombola* territories, and our small sample of 100 *Quilombola* territories (the highest number available from the Imaflora dataset in the Atlantic Forest with data for all our covariates) may not have been representative of the other 5,972 *Quilombola* territories across Brazil. If the Brazilian government were to register the thousands of *Quilombola* territories that have not been able to start the process of formalizing land claims, then there would more data available on *Quilombola* territories, it would be possible to better understand land use trends in *Quilombola* territories, and there would be a clearer path forward for *Quilombola* territories to obtain land rights”

I (the lead author on this paper) have another publication that is currently on its third round of review and will be published imminently, which highlights the problem of lack of available data on QTs and suggests a few possible paths forward for obtaining this data.

• Table S1 – I would find it really helpful to have information on how these different covariates vary by land use tenure. This would help to put the results of the study in context. For example, on lines 298-299 the authors argue that protected areas may not have contributed much in terms of restoration success because they may have already been highly forested, leaving little space for forest restoration. Surely this could be confirmed by examining the covariates, couldn't it?

We have now added an additional figure into the SI (Fig. S3–see below) that displays the variation of each covariate. This figure confirms the hypothesis that many protected areas were highly forested in the year 1985, as we had already mentioned in the Discussion. We now refer to this figure in the Discussion.

Figure S1. Distributions of covariate variation by land tenure regime.

Minor issues

- Table 1 – The information in this table is vital to understand the context of the work, but is there a way to display the information in a figure rather than a table?

We have now added a new Figure into the SI (Fig. S4) that visualizes the final two columns of Table 1 (see below). However, given the long sentences included in Table 1, we believe that a table is the most effective way to communicate this information, and have decided to keep Table 1 in the manuscript.

Figure S4. Number of lands and total land area of land tenure regimes.

• Table 2 – Similar to my comment above, could this table be turned into a figure? You could potentially take the information from the table and produce two separate plots for restoration successes and restoration reversals. I feel like this would help to communicate the message of the study much more effectively.

We have now replaced the table with a figure (Fig. 1–see below) to display our main results. We have now moved what was previously Table 2 into the SI.

- Lines 164 – 363: I recommend not using many acronyms in the discussion as it makes it difficult to follow what is being said.

We have now changed the acronyms to the full phrases throughout the entire manuscript for clarity.

- Lines 423 – 425: I would appreciate more detail about why the covariates selected were used.

We have now added an additional sentence in the Methods to explain why the covariates selected were used. The section now reads:

Lines 485-488: “Variables were selected based on data availability and precedence from prior studies^{19,30,34,35,92}. Specifically, a list of possible covariates was compiled from a literature review of similar studies, and covariates were included in the analysis that were possible to collect through publicly-available data sources.”

- Lines 435 – 502: There is no mention of the programs used for analyses aside from Google Earth Engine.

We have now added the phrase “in R” after mentioning our use of the MatchIt package and margineffects package to specify that we utilized these packages with R software.

References

1. Wauchop, H. S. et al. Protected areas have a mixed impact on waterbirds, but management helps. *Nature* 605, 103–107 (2022).
2. Geldmann, J., Manica, A., Burgess, N. D., Coad, L. & Balmford, A. A global-level assessment of the effectiveness of protected areas at resisting anthropogenic pressures. *Proc. Natl. Acad. Sci. U. S. A.* 116, 23209–23215 (2019).
3. Oldekop, J. A., Holmes, G., Harris, W. E. & Evans, K. L. A global assessment of the social and conservation outcomes of protected areas: Social and Conservation Impacts of Protected Areas. *Conserv. Biol.* 30, 133–141 (2016).
4. Hajjar, R. et al. A global analysis of the social and environmental outcomes of community forests. *Nat. Sustain.* 4, 216–224 (2020).
5. Nakagawa, S. & Freckleton, R. P. Missing inaction: the dangers of ignoring missing data. *Trends Ecol. Evol.* 23, 592–596 (2008).

Reviewer #3 (Remarks to the Author):

Key results

The paper demonstrates that Indigenous lands (ILs), and agrarian reform settlements (ARSs) had more long-term restoration successes than Private Properties (PPs) when controlling for biophysical conditions. ILs and ARSs concurrently had the higher rates of restoration reversal compared to PPs. Combined these

results indicate that avoiding land tenure transition to PPs could improve the efficiency of long-term restoration outcomes.

From a methodological point of view the paper presents a new and innovative technique of agglomerative matching that iteratively combines multiple PPs into private agglomerates to achieve covariate balance when matching treatment and control territories. This enables to avoid dropping any data retaining the analysis of 1.9 million territories in The Atlantic Forest (AF) biome that is a top priority ecosystem for restoration.

Validity

Data are gathered from a representative sample (1.9 million territories) and are described as the most reliable in terms of what is currently available (e.g. Imaflora, Table S1). Methods are described in a transparent and reproducible way with limitations highlighted. Each model was run 500 times to compare standardized mean differences (SMD) and prognostic scores across iterations.

Significance

The implications of land tenure regimes for restoration are explained in context. The broader applications of the results for restoration decision-makers and practitioners are presented in a concise and effective way.

Data and methodology, and analytical approach

The validity of the approach looks sound, and the methodology is presented in an easy-to-follow way. I can understand the matching approach, the algorithm, analysis and balance. I also assessed the data provided in the supplementary material and they look robust. However, I am not an expert in matching techniques. The code needs to be checked by someone more expert (e.g. Table S2, Figure S1 and S2).

We are pleased to hear that you believe the validity of the approach looks sound and the methodology was presented in an easy-to-follow way.

Suggested improvements, clarity and context

Abstract

In general, this could be made more effective by removing the reference to 189 ha and 21 ha. At first one doesn't understand why only a value for ILs is reported and not two (ILs and ARSs) since both are mentioned. So, my suggestion would be to just remove the values or alternatively provide both, but this could make the abstract too long.

We have now removed the suggested values from the abstract.

Also, I would suggest to stress that ILs and ARSs concurrently had the higher rates of success and restoration reversal compared to PPs. How it is phrased at the moment, it almost looks like a contradiction. So, I would add 'concurrently or in combination' to reassure the reader that this is indeed not a contradiction. Specify that the total area of restoration successes was around ten times more than reversals.

Perhaps briefly explain the reasons too of this concurrent results (success and restoration reversals on the same type of land tenure).

We have now added the phrase “Concurrently, and by a smaller margin and on a smaller land area” to the beginning of the suggested sentence. This change adds the word you suggested (concurrently) into the sentence and emphasizes that the size of the land area of reversals is smaller than successes. We were not able to add a longer explanation due to the abstract word limit.

The last sentence is too broad. I would suggest writing something more specific along the lines of ... Combined these results indicate that avoiding land tenure transition to PPs could improve the efficiency of long-term restoration outcomes.

**We have now replaced the previous final sentence of the abstract to make it more specific. We believe that the most important policy implications relate to 1) improving the longevity of restoration in private properties, and 2) recognizing land rights and supporting restoration in land tenure regimes with high rates of restoration successes. As such, the new sentence reads:
Lines 18-20: “These results suggest relatively low restoration longevity in private properties, and opportunities for restoration initiatives to recognize and support the socio-political conditions that enable long-term restoration success.”**

Lines 14-20 to be rephrased in particular

[... We find that ILs and ARSs have significantly more long-term restoration successes than PPs (189 ha more per IL than PPs). Restoration reversals are also significantly higher in ILs and ARSs, although by a smaller margin (21 ha more restoration reversals per IL than PPs). By characterizing how restoration reversals and successes vary between land tenure regimes, our study may support efforts to improve the longevity of forest restoration].

We have now rephrased the suggested sentences (see above).

Introduction

Lines 32-36 Can this be rephrased in a clearer way? As it is further down in the paper?

When evaluating forest restoration (hereafter restoration), one critical distinction is the difference between restoration successes—restored forests that remain forested for the long term (defined here as standing forests that have persisted for a minimum of 10 years and continued to persist), and restoration reversals—restored forests that are later deforested (defined here as persisting between 3-10 years—see Methods).

Here is much clearer ... We defined restoration success as restoration that persisted until the end of the time period (2022), and that persisted for a minimum of ten years. We defined restoration reversal as restoration that persisted between 3-10 years and was deforested before the end of the time period

**We have now added the same wording to the Introduction that was previously used later in the manuscript to clarify the definitions of restoration reversals and successes. The section now reads:
Lines 32-38: “When evaluating forest restoration (hereafter *restoration*)—defined as forest gain, including any change from productive land uses to natural forest cover—one critical distinction is the difference between *restoration successes*—restored forests that remain forested for the long term (defined here as restoration that persisted for a minimum of ten years and until the end of the measured time period (2022)), and *restoration reversals*—restored forests that are later deforested**

(defined here as restoration that persisted between 3-10 years and was deforested before the end of the measured time period—see Methods).”

Results

Lines 155-161 Descriptive statistics of restoration reversals and successes

I would move this at the beginning of the results

The main results of our paper are the results of the causal analysis. Accordingly, we would prefer to list the results of our main analysis first, and then to present the descriptive statistics afterwards. However, if you and the Editor believe this would be a useful change, we would be happy to change the order of how our results are presented.

References

The results are provided with sufficient context and consideration of previous work. References to previous literature are presented appropriately.

We are pleased to hear that you believe our results have been presented sufficiently and the references are used appropriately.

We thank the three reviewers for their careful consideration of our manuscript. We are grateful for their comments and have now incorporated their suggestions. **Our responses to each comment are shown in bold below.**

REVIEWERS' COMMENTS

Reviewer #1 (Remarks to the Author):

1. What are the noteworthy results?

The authors have significantly improved the manuscript and addressed all the points raised by the reviewers. The work now contributes to the understanding of land use changes in the Brazilian Mata Atlantica. The main result is the difference between restoration gains in indigenous lands compared with other tenure regimes.

Thank you for your review. We are pleased to hear that we have addressed all the points raised by the reviewers.

2. Are there any flaws in the data analysis, interpretation and conclusions?

The revised version improved the data analysis, and led to conclusions which are tenable. The only concern is the use of the term "restoration success", which implies a deliberate action by the tenant to achieve a goal. As the authors note, they make no distinction between areas which have resulted from a restoration project and areas which have been abandoned. For this reason, the authors are requested to use a more neutral term, such as "restoration gain", instead of "restoration success".

We have now replaced the term 'restoration success' with the new more neutral term 'restoration gain' throughout the manuscript.

3. Is the methodology sound? Does the work meet the expected standards in your field?

The methods appear to be sound, based on the results. However, the "Methods" section still needs a revision. The sentence "Our agglomerative matching algorithm iteratively sorts through private properties (see algorithm below) to select a combination of private properties with covariates similar to each treatment territory" introduces the concept of "treatment territory" that is neither described in the main text nor in SI. Please explain this term and use a more English-sounding concept. Would "spatial cluster" be more adequate. After all, what the authors are doing is to join similar PPs in a spatial cluster.

We have now added an additional sentence to define the terms 'treatment land' and 'control land'. The section now reads: "Our *agglomerative matching* algorithm iteratively sorts through private properties (see algorithm below) to select a combination of private properties with covariates similar to each Indigenous land, agrarian-reform settlement, *Quilombola* territory, and protected

area. Treatment lands were Indigenous lands, agrarian-reform settlements, *Quilombola* territories, or protected areas, and control lands were private properties.”

Additional changes to the methods:

-We have now slightly softened our causal language in three instances in the Abstract and one instance at the end of the Introduction to more robustly describe our matching technique.

4. Is there enough detail provided in the methods for the work to be reproduced?

Yes, there is enough detail to reproduce the paper.

Overall recommendation: ACCEPT WITH MINOR REVIEWS

Reviewer #1 (Remarks on code availability):

The code is clear and easy for an R expert to understand.

Reviewer #2 (Remarks to the Author):

I am satisfied that the authors have addressed all the points that I raised in the previous round of review and that the manuscript is more robust as a result. Good work!

After reading the manuscript I found no issues with the current version.

We are very pleased to hear that we have addressed all of your points and that you have no issues with the current version.

Reviewer #3 (Remarks to the Author):

I have reviewed the revised version of the manuscript and find that the authors have addressed all of my previous comments comprehensively and to a high standard. The revisions have improved the clarity and rigor of the work, and I am satisfied with the responses provided.

While I have not reviewed the accompanying code, based on the manuscript and the authors' detailed responses, I have no further concerns, and I support the publication.

Best regards

We appreciate your review and are glad that we have addressed all of your previous comments.